# Diabetes Mellitus as a Predictive Factor for Urinary Tract Infection for Patients Treated with Kidney Transplantation

**DOI:** 10.3390/medicina58101488

**Published:** 2022-10-19

**Authors:** Kaori Ozawa, Manabu Takai, Tomoki Taniguchi, Makoto Kawase, Shinichi Takeuchi, Kota Kawase, Daiki Kato, Koji Iinuma, Keita Nakane, Takuya Koie

**Affiliations:** 1Department of Urology, Ogaki Municipal Hospital, Ogaki 5038502, Japan; 2Department of Urology, Gifu University Graduate School of Medicine, Gifu 5011194, Japan

**Keywords:** kidney transplantation, diabetes mellitus, urinary tract infection

## Abstract

*Background and Objectives*: We aimed to investigate the rate of incidence and risk factors of post-transplant urinary tract infection (UTI) in patients receiving kidney transplantation (KT) at our institution. *Materials and Methods*: A retrospective cohort study was carried out on patients who underwent KT for end-stage kidney disease (ESKD) from January 2008 to December 2021 at Gifu University Hospital. UTI was defined as the existence of bacterial and/or fungal infection in urine with ≥105 colony-forming units/mL, with or without urinary and/or systemic symptoms of UTI. Patients were divided into two groups: those with UTI after KT (UTI group) and those without UTI (non-UTI group). The primary endpoint of this study was the relationship between covariates and UTI after KT. *Results*: Two hundred and forty patients with ESKD received KT at Gifu University Hospital. Thirty-four participants developed UTI after surgery, and the most common pathogen was *Escherichia coli*. At the end of the follow-up, graft loss was observed in six patients (2.5%), independent of UTI episodes. In the multivariate analysis, diabetes mellitus (DM) was statistically associated with post-transplant UTI in kidney transplant recipients. *Conclusions*: Preoperative serum glucose control in patients with DM may have a crucial role in preventing UTI and preserving renal function after KT.

## 1. Introduction

For patients with end-stage kidney disease (ESKD) receiving dialysis, kidney transplantation (KT) may have potential benefits in increasing the quality of life and decreasing the risk of mortality [1]. However, the risk of infection in recipients with KT may be high compared with that in the general public because of the continuous administration of intensive immunosuppressive drugs, and the main cause of mortality and morbidity is contagious bacterial disease [2,3]. Urinary tract infections (UTI) are the most common infectious diseases for KT patients—from 20 to 80% of KT patients develop UTIs [4,5]. Various risk factors have been distinguished which increase the risk of UTI after KT [2,6,7,8]. These factors include older age, female sex, diabetes mellitus (DM), history of acute injection, delayed graft function, deceased donor KT, longer duration of dialysis, urological abnormalities, and timing of stent removal [2,6,7,8]. Furthermore, the type of immunosuppression is strongly associated with the development of UTI [9]. It is well-known that immunosuppression affects the resistance of enterococci to β-lactam-based antibiotics and the expression of penicillin-binding proteins [9]. Azathioprine, mycophenolate mofetil (MFF), and anti-thymocyte globulin have a higher incidence of UTIs after kidney transplantation, while calcineurin inhibitors and everolimus do not seem to affect the risk of urinary tract infection [9].

UTI is the most common infectious disorder in patients with DM, and 150 million patients worldwide suffer from UTI each year [10]. UTI in patients with DM is caused by immune system disorders, decreased white blood cell function, poor blood supply, bladder dysfunction due to neuropathy, and glucosuria [11]. Although pre- and post-transplantation DM have been significantly correlated with an increased incidence of mortality and infection in recipients who underwent parenchymatous organ transplantation, including KT [12], the association between UTI and DM remains unclear. Therefore, we investigated the rate and risk factors of postoperative UTI in patients who underwent KT at our institution.

## 2. Materials and Methods

### 2.1. Patients

This study was approved by the Institutional Review Board of Gifu University (authorization number: 2021-B129, approved on 15 November 2021). The requirement for obtaining informed consent was waived due to the retrospective nature of the study. The provisions of the Ethics Committee and ethics guidelines in Japan did not require written consent because the study information was disclosed to the public, as is the case of retrospective and observational studies that use materials such as existing documentation. The details of this study can be accessed at https://www.med.gifu-u.ac.jp/visitors/disclosure/docs/2021-B129.pdf (accessed on 6 June 2022).

This retrospective cohort study included patients with ESKD who received KT from January 2008 to December 2021 at Gifu University Hospital. Preoperative information included patient age, sex, dialysis, dialysis vintage, primary renal disease of the recipients, donor type, cytomegalovirus (CMV) infection, rejection after KT, ureteral stent indwelling, and removal time of the ureteral stent and urethral catheter. Additionally, the patients were divided into two groups: those with UTI after KT (UTI group) and those without UTI (non-UTI group).

### 2.2. Immunosuppression Regimen

The enrolled recipients administered an induction treatment with immunosuppressive drugs using an anti-interleukin-2 antibody (basiliximab) and maintenance triple immunosuppressive drugs as the standard therapy, including tacrolimus, MMF, and methylprednisone, according to our institutional protocol [13]. In the current study, enrolled kidney transplant recipients (KTRs) who performed blood type incompatible KT were administered rituximab before KT. Rituximab was administered before KT to high-risk recipients who were highly sensitized to a positive donor-specific antibody or negative serological crossmatch and underwent ABO-incompatible KT [12].

### 2.3. The Timing of the Removal of Ureteral Stent and Urethral Catheter

Double-J stents and urethral catheters were placed in all recipients at our institution. If the recipients progressed favorably after KT, both catheters were removed on postoperative day eight.

### 2.4. Antimicrobial Prophylaxis

All patients who underwent KT were routinely administered prophylactic anti-CMV treatment with valganciclovir for 3 months with respect to the immunoglobulin G status of the donor and/or recipient, and trimethoprim-sulfamethoxazole prophylaxis as the antibiotic against *Pneumocystis jirovecii* was used at 80/400 mg/day, three times a week, for 6 months after KT. As perioperative antimicrobials for recipients, cephazolin sodium was used for 2 days after KT according to our institutional protocol. None of the patients who underwent KT routinely received additional antibacterial prophylaxis at our institution.

### 2.5. The Definition of UTI

Urine analyses and cultures were performed at each consultation after KT. UTI was defined as the existence of bacterial and/or fungal infection in urine with ≥10^5^ colony-forming units/mL with or without urinary and/or systemic symptoms of UTI [14]. Asymptomatic bacteriuria was defined as the absence of symptoms associated with UTI and a positive urine culture.

Multidrug-resistant microorganisms (MDROs) have been identified as extended-spectrum β-lactamase-producing bacteria, such as methicillin-resistant Staphylococcus aureus, Acinetobacter baumannii, Enterococcus faecium, and *Pseudomonas aeruginosa*, which are tolerant to at least one of three or more antimicrobial agents [15].

### 2.6. Statistical Analyses

The primary endpoint of this study was the association between covariates and UTI after KT. The software JMP 14 (SAS Institute Inc., Cary, NC, USA) was used for data analysis. The differences between patients with and without UTI after KT were compared using the Mann–Whitney U test for categorical variables and the chi-square or Fisher’s exact test for continuous variables. Graft survival (GS) was evaluated using the Kaplan–Meier method. The association between GS and subgroup classification was analyzed using the log-rank test. GS was determined as the interval from KT to the loss of a transplant kidney or death. Multivariate analysis was performed using a Cox proportional hazards model according to the predictive factors of UTI after KT. All *p*-values were two-sided, and statistical significance was set at *p* < 0.05.

## 3. Results

### 3.1. Patients

A total of 236 patients with ESKD performed KT at Gifu University Hospital. Four recipients were excluded from this study because they were ≤18 years old. Patient characteristics are shown in Table 1. Postoperative UTI occurred in 33 (14.0%) patients. In the UTI group, 20 patients (60.6%) had symptomatic UTIs. The median age, dialysis vintage, and follow-up period in the enrolled patients were 47 years (interquartile range (IQR), 38–59 years), 8 months (IQR, 0–34 months), and 160.8 months (IQR, 40.3–117.0 months), respectively. Eighty-three patients (35.2%) underwent KT as preemptive transplantation, and 22 (9.3%) transplants were from deceased donors. A total of 34 (14.4%) patients had diabetic nephropathy due to preoperative DM. The ureteral stent was indwelling in 231 patients (97.9%), and the median removal time of the ureteral stent was 8 days (IQR, 7–10 days). The median time for the urethral catheter removal was 8 days (IQR, 7–9 days) after KT in the enrolled patients.

### 3.2. Microbiological Results and Risk Factors for UTI

The postoperative pathogens from the urine samples of patients with KT at the outpatient clinic are shown in Table 2. *Escherichia coli* was the most common pathogen. In total, four isolates (11.8%) were identified as MDROs.

At the end of the follow-up, graft loss was observed in six patients (2.5%), independent of UTI episodes. The 5-year graft survival rate was 98.6% and 100% in recipients without and with UTI, respectively (*p* = 0.895; Figure 1).

In the multivariate analysis, DM was significantly associated with a predictive factor for postoperative UTI in patients treated with KT (Table 3). The post-transplant period was not considered a factor in the multivariate analysis because the dosage of immunosuppressive drugs in use and the degree of immunosuppression varied from patient to patient.

## 4. Discussion

KT is a useful treatment modality in patients with ESKD. Post-transplant infection, especially UTI, is more likely to develop as a postoperative complication, despite remarkable progress in surgical techniques and immunosuppressive medicines [7]. The risk of death from infection is 32 times higher for those who received KT, and mortality due to sepsis is 20-fold higher than that in the general public [3,16]. Several studies have reported that postoperative UTIs in patients undergoing KT are important causes of acute graft dysfunction if not adequately treated [2,7,14,17]. In addition, recipients who developed rejection are more prone to UTI, and recurrent UTIs in KT recipients are associated with an increased risk of fibrosis of the renal allograft [7,18]. Powers et al. investigated that rejection was a significant risk factor for histological acute graft pyelonephritis (HAGPN); additionally, 50% of patients who underwent KT had concomitant clinically significant rejection, although it is still unclear how acute rejection is involved in the development of HAGPN [19]. Although bacterial colonization of the urinary tract may be a mechanism for renal allograft infection, almost half of patients with HAGPN lack significant pathogen growth [19,20]. It is speculated that bacterial infections may activate both innate and adaptive immunity, increasing the risk of rejection [21].

In the current study, DM was statistically detected for predicting post-transplant UTI based on multivariate analysis. Patients with DM generally have a potential disadvantage of an increased risk of infections, especially UTIs [11]. According to a retrospective case observational study from the UK General Practice Research Database, the incidence of UTI was 46.9 per 1000 person-years (95% confidence interval (CI), 45.8–48.1) in recipients with DM and 29.9 (95% CI, 28.9–30.8) in those without DM, and the risk of developing a UTI was higher in patients with DM than in those without DM [22]. Various mechanisms are thought to increase the risk of UTI, including diabetic nephropathy, autonomic neuropathy, immune system disorders, and glucosuria [11]. Among them, diabetic nephropathy leads to protein excretion, severe glucosuria, and decreased local cytokine secretion [11,23]. In patients with DM, there is a risk of progression of the infection to a more severe stage of UTI or the development of emphysematous cystitis and pyelonephritis [24]. Gram-negative bacteria such as *Escherichia coli* and *Klebsiella pneumoniae* are the most commonly isolated Gram-negative bacteria from UTI specimens from KT patients; however, some studies have shown an increase in infections caused by MDROs. The rate of infections by MDROs ranged from 6.5% to 56% in solid organ recipients [16,25]. Ramadas et al. reported that decreased local immunity might increase the risk of infection by MRDOs [25]. Nevertheless, maintaining control of serum glucose may play an important role in avoiding UTI and preserving renal function after KT [24].

Prevention of UTI after KT may be enhanced by early removal of the ureteral stent [4]. Ureteral stents through the ureterovesical anastomosis can reduce urologic complications such as urinary leakage, ureteral necrosis, and anastomotic stenosis or obstruction. A recent systematic review and meta-analysis, including 14 studies with a total of 3216 patients with KT, showed that removal of the stent earlier than 3 weeks without an increased incidence of leakage compared with delayed removal after 3 weeks (odds ratio (OR), 0.49; 95% CI, 0.33–0.75, *p* < 0.001) indicated a significant reduction in UTI [26]. Several studies have also reported that early removal of ureteral stents after KT may reduce the incidence of UTI without significantly increasing major urological complications [4,27,28]. The guidelines from the Infectious Disease Community of Practice of the American Society of Transplantation recommend that the duration of catheter and stent placement be within 4 weeks of KT in patients who undergo KT [4]. In our study, both the ureteral stent and urethral catheter were removed at a median of 8 days after KT, without surgery-related complications. Therefore, removal of the ureteral stent and urethral catheter with the least possible delay may decrease the risk of UTIs after KT.

This study has certain limitations. First, because this is a retrospective study conducted at a single institution, the possibility of selection bias cannot be ruled out. Second, the study had a relatively small sample size and a relatively short follow-up period. In this study, UTI was defined as a positive urine culture. Therefore, we did not investigate the association between UTI and clinical symptoms. Additionally, we did not evaluate the statistical differences between patients who had positive urine cultures without clinical symptoms and those who had UTI-related urinary and/or systemic symptoms. Finally, post-transplant DM was not evaluated in this study.

## 5. Conclusions

KT recipients may be at higher risk of UTI after KT. Our study identified DM as an independent predictive factor for an increased risk of UTI after KT. The prevalence of DM has increased, and the number of KT recipients with DM is expected to increase in the near future. Preoperative serum glucose control may play an important role in preventing UTI and preserving renal function after KT. Further prospective, large-scale, long-term studies are needed to validate these results.

## Figures and Tables

**Figure 1 medicina-58-01488-f001:**
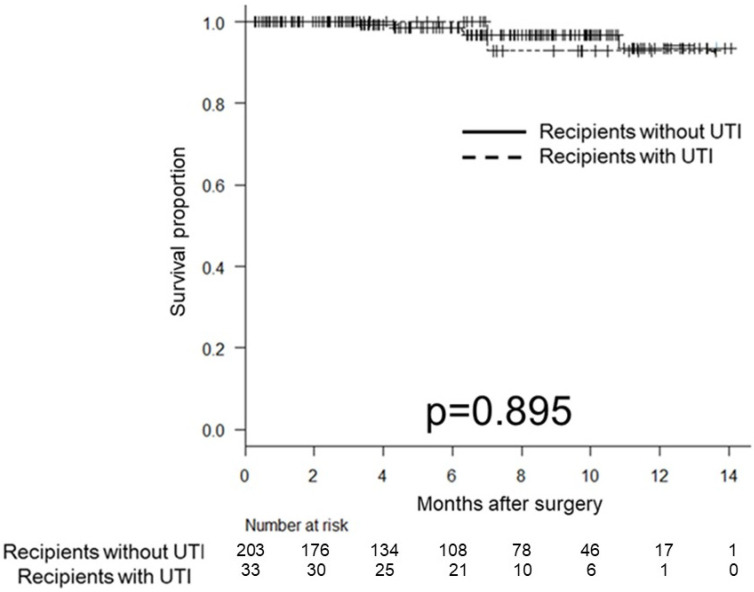
Kaplan–Meier estimates of graft survival according to urinary tract infection (UTI) after kidney transplantation (KT); the 5-year graft survival was 98.6% in KT patients without UTI and 100% in KT patients with UTI (*p* = 0.895).

**Table 1 medicina-58-01488-t001:** Clinical characteristics of patients.

Covariates	UTI	Non-UTI	*p*
Number	33	203	
Age (years, median, IQR)	51 (45–59)	47 (38–59)	0.169
Sex (number, %)			0.130
Male	16 (48.5)	127 (62.6)	
Female	17 (51.5)	76 (37.4)	
Dialysis (number, %)			0.479
Hemodialysis	18 (54.5)	116 (57.1)	
Peritoneal dialysis	2 (6.1)	27 (13.3)	
Preemptive transplantation (number, %)	13 (39.4)	60 (29.6)	0.310
Dialysis vintage (months, median, IQR)	9 (0–54)	8 (0–33)	0.939
Diabetes mellitus (number, %)	9 (27.3)	25 (12.3)	0.032
Recipient renal disease (number, %)			0.056
Primary glomerulonephritis	11 (33.3)	107 (52.7)	
Diabetic nephropathy	9 (27.3)	25 (12.3)	
Renal urological diseases	4 (12.1)	25 (12.3)	
Hypertensive nephropathy	2 (6.1)	10 (4.9)	
Genetic disorder	0	4 (2.0)	
Interstitial nephritis	2 (6.1)	1 (0.5)	
Others	5 (15.2)	31 (15.3)	
Donor type (number, %)			0.748
Deceased	2 (6.1)	20 (9.9)	
Living, related	31 (93.9)	183 (90.1)	
Rejection after renal transplantation (number, %)	0	6 (3.0)	>0.999
Ureteral stent replacement (number, %)	32 (97.0)	199 (98.0)	0.532
Ureteral stent removal time (days, median, IQR)	9 (7–14)	8 (7–9)	0.305
Urethral catheter removal time (days, median, IQR)	8 (7–11)	8 (7–9)	0.972

UTI, urinary tract infection; IQR, interquartile range.

**Table 2 medicina-58-01488-t002:** Bacterial isolates from urine in patients with urinary tract infections.

Microorganism Species	Number (%)
*Escherichia coli*	21 (60.0)
*Pseudomonas aeruginosa*	5 (14.2)
*Enterococcus faecalis*	3 (8.5)
*Klebsiella pneumoniae*	2 (5.7)
*Aerococcus urinae*	1 (2.9)
*Staphylococcus epidermidis MRSE*	1 (2.9)
*Streptococcus agalactiae*	1 (2.9)
*Ralstonia pickettii*	1 (2.9)

MRSE, methicillin-resistant *Staphylococcus epidermidis*.

**Table 3 medicina-58-01488-t003:** Multivariate analysis for urinary tract infection after kidney transplantation.

Risk Factor	Odds Ratio	95% Confidence Interval	*p*
Diabetes mellitus	0.39	0.16–0.93	0.034
Age	1.03	0.99–1.06	0.097
Ureteral stent removal time	1.05	0.94–1.18	0.369
Urinary catheter removal time	0.95	0.85–1.07	0.418
Dialysis	0.79	0.36–1.72	0.556

## Data Availability

Data and materials are provided in this paper.

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
