# Peer review of "Diabetes Mellitus as a Predictive Factor for Urinary Tract Infection for Patients Treated with Kidney Transplantation"

_medicina, 2022, doi:10.3390/medicina58101488_

Round 1
Reviewer 1 Report
Minor spellcheck:
line 21 and 98: UIT --> UTI
line 45: I believe bladder dysfunction due to neuropathy instead of nephropathy?
line 45: quotation [10]
line 56: written
line 78: I believe all instead of almost
Please avoid double meanings like line 100 KT patients with and without UTI after KT to improve fluid reading
Minor revision:
- asymptomatic bactiuria?
- when was screened for UTI? How long? Only during the hospital stay?
- cox regression done with consideration of immunosuppression that is very high in the beginning and lower in the course of time?
Major revision:
Numbers in results text and table dont match.
line 112-114 respectively? Please name the groups.
Please discuss the influence of immunosuppression on UTI and insert the point that you highlight in the introduction in your analysis.
How long was the follow-up? How long was the observation period?
Please insert PMID: 30689126 as a reference
Author Response
15, October, 2022
Dr. Editor
The Medicina
Dear Editor:
Thank you very much for the review of our manuscript titled “Diabetes mellitus as a predictive factor for urinary tract infection for patients treated with kidney transplantation.”
We sincerely appreciate all valuable comments and suggestions, which helped us to improve the quality of our manuscript. Our responses to the Reviewers’ comments are described below in a point-to-point manner. Appropriate changes, suggested by the Reviewers, have been introduced to the manuscript (track-changes mode in the red color font). Let me emphasize our full readiness to make any further improvements to the manuscript.
We hope that our manuscript will be acceptable for publication in the Medicina.
We look forward to hearing from you.
Yours sincerely,
Takuya Koie
Corresponding author
Department of Urology
Gifu University Graduate School of Medicine
1-1 Yanagido, Gifu, Gifu 501-1194, Japan
TEL.: +81-582-30-6338
FAX: +81-582-30-6341
e-mail: goodwin@gifu-u.ac.jp
Responses to the reviewer's comments
We would like to thank the Reviewers for taking the time and effort necessary to review the manuscript. We sincerely appreciate all the valuable comments and suggestions, which helped us to improve the quality of the manuscript.
Response to Reviewer 1
The authors appreciate the Academic Editor’s comments. The authors’ point-by-point responses to the comments are given below.
1) line 21 and 98: UIT --> UTI.
Response:
The authors have revised the following part on line 21:
and UIT UTI after KT.
The authors have revised the following part on line 106:
and UIT UTI after KT.
2) line 45: I believe bladder dysfunction due to neuropathy instead of nephropathy?
line 45: quotation [10]
Response:
The authors have revised the following part on line 51:
due to nephropathy, neuropathy and glucosuria.10 [11].
3) line 56: written
Response:
The authors have revised the following part on line 62:
require writ-ten written consent
4) line 78: I believe all instead of almost
Response:
The authors have revised the following part on line 84:
placed in almost all recipients
5) Please avoid double meanings like line 100 KT patients with and without UTI after KT to improve fluid reading
Response:
The authors have deleted the following word on line 108:
KT patients with and without UTI after KT
6) - asymptomatic bactiuria?
Response:
The authors have added the following sentence on line 93:
Asymptomatic bacteriuria was defined as the absence of symptoms associated with UTI and a positive urine culture.
7) when was screened for UTI? How long? Only during the hospital stay?
Response:
The authors have already described the following sentence on line 96:
Urine analyses and culture were performed at each consultation after KT.
8) - cox regression done with consideration of immunosuppression that is very high in the beginning and lower in the course of time?
Response:
The authors have added the following sentence on line 146:
The post-transplant period was not considered as a factor in the multivariate analysis because the dosage of immunosuppressive drugs in use and the degree of immunosuppression varied from patient to patient.
9) Numbers in results text and table do not match.
Response:
The authors have revised the following sentence on line 118:
A total of 236 240 patients with ESKD performed KT at the Gifu University Hospital.
The authors have revised the following sentence on line 120:
Postoperative UTI occurred in 33 34 (14.0 14.4%).
The authors have revise patients at risk in Figure 1.
10) line 112-114 respectively? Please name the groups.
Response:
The authors have added the following sentences on line 121:
The median age, dialysis vintage, and follow-up period in the enrolled patients were 47 years
11) Please discuss the influence of immunosuppression on UTI and insert the point that you highlight in the introduction in your analysis.
Response:
The authors have added the following sentences on line 41:
Furthermore, the type of immunosuppression is strongly associated the development of UTI [9]. It is well known that immunosuppression affects the resistance of enterococci to β-lactam-based antibiotics and the expression of penicillin-binding proteins [9]. Azathioprine, mycophenolate mofetil (MFF), and anti-thymocyte globulin have a higher incidence of UTIs after kidney transplantation, while calcineurin inhibitors and everolimus do not seem to affect the risk of urinary tract infection [9].
12) How long was the follow-up? How long was the observation period?
Response:
The authors have already described the following sentence on line 121:
The median age, dialysis vintage, and follow-up period in the enrolled patients were 47 years (interquartile range [IQR], 38–59 years), 8 months (IQR, 0–34 months), and 160.8 months (IQR, 40.3–117.0 months), respectively.
13) Please insert PMID: 30689126 as a reference
Response:
The authors have quoted this manuscript as ref 9.

Reviewer 2 Report
Concerns: Retrospective design. Long term recruitment period. Could be affected by changes in protocols. Asymptomatic urinary infection was included as outcome. Did the authors performed specific analysis for asymptomatic versus symptomatic UTI?.
Suggestion: Please provide information about lymphocuye counts in both groups or IgG levels if available. If a significative difference is observed please include as a risk factor in multivariate analysis.
Author Response
15, October, 2022
Dr. Editor
The Medicina
Dear Editor:
Thank you very much for the review of our manuscript titled “Diabetes mellitus as a predictive factor for urinary tract infection for patients treated with kidney transplantation.”
We sincerely appreciate all valuable comments and suggestions, which helped us to improve the quality of our manuscript. Our responses to the Reviewers’ comments are described below in a point-to-point manner. Appropriate changes, suggested by the Reviewers, have been introduced to the manuscript (track-changes mode in the red color font). Let me emphasize our full readiness to make any further improvements to the manuscript.
We hope that our manuscript will be acceptable for publication in the Medicina.
We look forward to hearing from you.
Yours sincerely,
Takuya Koie
Corresponding author
Department of Urology
Gifu University Graduate School of Medicine
1-1 Yanagido, Gifu, Gifu 501-1194, Japan
TEL.: +81-582-30-6338
FAX: +81-582-30-6341
e-mail: goodwin@gifu-u.ac.jp
Responses to the reviewer's comments
We would like to thank the Reviewers for taking the time and effort necessary to review the manuscript. We sincerely appreciate all the valuable comments and suggestions, which helped us to improve the quality of the manuscript.
Response to Reviewer 2
The authors appreciate the Academic Editor’s comments. The authors’ point-by-point responses to the comments are given below.
Concerns: Retrospective design. Long term recruitment period. Could be affected by changes in protocols. Asymptomatic urinary infection was included as outcome. Did the authors performed specific analysis for asymptomatic versus symptomatic UTI?
Response:
The authors have added the following sentence on line 120:
In the UTI group, 20 patients (60.6%) had symptomatic UTI.
Suggestion: Please provide information about lymphocyte counts in both groups or IgG levels if available. If a significative difference is observed please include as a risk factor in multivariate analysis.
Thank you for your great comments. However, this study did not investigate factors associated with inflammation such as neutrophils, lymphocytes, or C-reactive protein. Unfortunately, we are unable to consider the factors you suggested.

Round 2
Reviewer 1 Report
line 65: link to further details about the study is in Japanese. Is there an English version available?
Reviewer 2 Report
The manuscript is more suitable for publication now.